# Logarithmic, noise-induced dynamics in the Anderson insulator

Talía L. M. Lezama[1] and Yevgeny Bar Lev[1]

[1]*Department of Physics, Ben-Gurion University of the Negev, Beer-Sheva 84105, Israel*

We study the dynamical behavior of the Anderson insulator in the presence of a local noise. We show that the noise induces logarithmically slow energy and entanglement growth, until the system reaches an infinite-temperature state, where both quantities saturate to extensive values. The saturation value of the entanglement entropy approaches the average entanglement entropy over all possible product states. At infinite temperature, we find that a density excitation spreads logarithmically with time, without any signs of asymptotic diffusive behavior. In addition, we provide a theoretical picture which qualitatively reproduces the phenomenology of particle transport.

## I. INTRODUCTION

Anderson localization is an ubiquitous wave phenomenon that arises due to destructive interference in the presence of quenched disorder [1]. Since its discovery, it has been instrumental for the understanding of a much richer class of physical phenomena [2–6]. One of the most important manifestations of Anderson localization is the suppression of transport which follows from the exponential localization of all the single-particle wavefunctions in one and two dimensions [1, 7], such that infinitesimally small disorder leads to zero DC-conductivity at any temperature. While in higher dimensions, and at zero temperature, a metal-insulator transition takes place [7]. The phenomenon of Anderson localization has been extensively studied [8, 9], and experimentally demonstrated in many systems [10–17]. Even if most of the experimental setups are highly controlled, it is impossible to completely isolate the system from all dissipation effects that arise due to the coupling to the environment, for example, coupling to phonons is present in any condensed matter system, and is known to induce a finite DC conductivity [18]. It is therefore of importance to theoretically account for such dissipative processes.

The stability of Anderson localization to different classes of perturbations has been assessed in several theoretical studies [4–6, 19–28]. Anderson localization is known to be stable under spatially local but quasiperiodic in time perturbations in any dimension [20], and to survive global periodic driving in one dimension [27, 29]. In contrast, it is unstable to global noise, which is known to induce delocalization [22, 23, 26, 28, 30–34], and a transient subdiffusive transport, which eventually crosses over to regular diffusion [26, 34]. Global noise was also shown to lead to prethermal energy plateaus at intermediate time scales, followed by exponential relaxation at longer time scales [28].

In this work, we study how the dynamics in the one-dimensional Anderson insulator is affected by the presence of a *local* white-noise, which can be thought as a coupling to a local Markovian bath. We find that the noise leads to a logarithmically slow heating of the system up to an infinite-temperature state which is further

reflected in slow transport properties of the system.

Our article is organized as follows. In Sec. II, we introduce the model and the methods used to characterize the noise-induced dynamics. In Sec. III, we first assess the heating dynamics in terms of the energy and the entanglement entropy. We then analyze the particle transport in the system using both numerical simulations and a semi-analytical approach. In Sec. V, we summarize and discuss our main results.

## II. MODEL AND METHODS

We consider the one-dimensional Anderson model,

$$\hat{H}_A = -J \sum_{i=1}^{L-1} \left( \hat{c}_i^\dagger \hat{c}_{i+1} + \text{h.c.} \right) + \sum_{i=1}^{L} w_i \hat{n}_i, \qquad (1)$$

where $\hat{c}_i^\dagger$ ($\hat{c}_i$) creates (annihilates) a spinless electron on site $i$, "h.c." stands for a Hermitian conjugate, $\hat{n}_i = \hat{c}_i^\dagger \hat{c}_i$ is the density, and $J$ denotes the hopping constant. The on-site disorder potential, $w_i$, are independent random variables uniformly distributed in the interval $w_i \in [-W, W]$, with $W$ the disorder strength. This model is known to be localized for any $W > 0$ [1]. We perturb the Anderson model, by the addition of a local white-noise,

$$\hat{H} = \hat{H}_A + \zeta(t)\hat{n}_{L/2}, \qquad (2)$$

such that $\zeta(t)$ has zero mean $\overline{\zeta(t)} = 0$ and a vanishing correlation length

$$\overline{\zeta(t)\zeta(t')} = \gamma\delta(t - t'), \qquad (3)$$

where the overbar denotes the average over stochastic realizations of the noise and $\gamma$ is the noise strength. The noise term represents a local Markovian heat bath coupled to the system, and as such, the dynamics of the density matrix of the system, $\hat{\rho}(t)$, is given by the following Lindblad equation [35]

$$\partial_t \hat{\rho}(t) = -i\left[\hat{H}_A, \hat{\rho}(t)\right] + \qquad (4)$$
$$+ \gamma\left(\hat{n}_{L/2}\hat{\rho}(t)\hat{n}_{L/2} - \frac{1}{2}\left\{\hat{n}_{L/2}, \hat{\rho}(t)\right\}\right),$$

where $\{\cdot, \cdot\}$ is the anti-commutator. The Lindblad equation describes a trace-preserving non-unitary evolution, where the first term corresponds to the unitary evolution, and the second term corresponds to the dissipative coupling between the system and the local heat bath. It is easy to check by substitution, that the steady state of (4) is an infinite-temperature state with density matrix $\hat{\rho}_\infty \propto \mathbb{1}$, which means that for any initial state the system will approach $\hat{\rho}_\infty$. Please note, that the approach to infinite temperature by itself does *not* imply delocalization of the system, since as stated above, in one and two dimensions, and without the coupling to the noise, localization persists also at infinite temperature [1]. Here, we focus on the questions of *how* the infinite-temperature state is approached and what is the dynamics of the system at this state, in the presence of a local noise.

While (4) can be numerically solved, this is extremely demanding even for noninteracting particles, since certain couplings to the heat bath create an effective interaction between the particles, which requires the use of the full many-body density matrix of dimensions $\mathcal{N} \times \mathcal{N}$, where $\mathcal{N}$ is the Hilbert-space dimension. Alternative methods, based on a unitary propagation followed by stochastic measurements, of an ensemble of wavefunctions, were developed [36–40]. These methods, known as quantum-trajectory methods, are more efficient since the dimension of the wavefunction is $\mathcal{N}$. The solution of (4) is reproduced by an average over quantum trajectories, which correspond to individual realizations of the measurements. The procedure of writing (4) as a stochastic differential equation, is known as "unraveling", and since there can be many stochastic differential equations whose averages reproduce (4), the procedure is not unique, and can depend on the physical context [38].

In this work, we use a unitary unraveling of (4), which was introduced in Refs. [39, 40] and corresponds to the following stochastic unitary infinitesimal propagator

$$\hat{U}\left(t + \mathrm{d}t, t\right) = e^{-i\hat{H}\mathrm{d}t - i\eta_t \hat{n}_{L/2}\sqrt{\gamma \mathrm{d}t}}, \tag{5}$$

where $\eta_t$ are independent normally distributed random variables. The evolution of the density matrix is then obtained by performing an average over trajectories corresponding to the different realizations of the noise $\eta_t$, namely,

$$\hat{\rho}\left(t + \mathrm{d}t\right) = \overline{\left|\psi\left(t + \mathrm{d}t\right)\right\rangle \left\langle \psi\left(t + \mathrm{d}t\right)\right|}, \tag{6}$$

where the overbar denotes the average over the noise trajectories, and $\left|\psi\left(t + \mathrm{d}t\right)\right\rangle = \hat{U}\left(t + \mathrm{d}t, t\right)\left|\psi\left(t\right)\right\rangle$, with the initial condition $\left|\psi\left(t = 0\right)\right\rangle$ taken from an ensemble whose average corresponds to the initial density matrix $\hat{\rho}(t = 0) = \overline{\left|\psi\left(t = 0\right)\right\rangle \left\langle \psi\left(t = 0\right)\right|}$.

For self-adjoint Lindblad operators this unraveling is equivalent to the quantum-jump approach [40], but is numerically superior for an initial quadratic density matrix

$\hat{\rho}\left(t = 0\right) = e^{-\sum_i \alpha_i \hat{n}_i}$, since it only requires the propagation of a *single-particle* density matrix

$$\rho_{ij}^s\left(t\right) \equiv \mathrm{Tr}\left(\hat{\rho}\left(t\right) \hat{c}_i^\dagger \hat{c}_j\right), \tag{7}$$

which is polynomial rather than exponential in $L$. This simplification occurs, since $\hat{U}\left(t + \mathrm{d}t, t\right)$ is quadratic in $\hat{c}_i^\dagger$ ($\hat{c}_i$) and therefore an initially quadratic density matrix, stays quadratic for the entire evolution of the system. The propagation of $\rho_{ij}^s\left(t\right)$ is obtained using the single-particle version of the stochastic unitary propagator (5),

$$U^s\left(t + \mathrm{d}t, t\right) = e^{-i\hat{h}_A \mathrm{d}t - i\eta_t |L/2\rangle\langle L/2|\sqrt{\gamma \mathrm{d}t}}, \tag{8}$$

where

$$\hat{h}_A = -J \sum_{i=1}^{L-1}\left(|i\rangle\langle i+1| + |i+1\rangle\langle i|\right) + \sum_{i=1}^{L} w_i |i\rangle\langle i| \tag{9}$$

is the single-particle Anderson Hamiltonian. The evolved single-particle density matrix $\rho_{ij}^s\left(t + \mathrm{d}t\right)$ is therefore given by,

$$\rho_{ij}^s\left(t + \mathrm{d}t\right) = \overline{U_{ik}^s\left(t + \mathrm{d}t, t\right) \rho_{kl}^s\left(t\right) \hat{U}_{jl}^{s*}\left(t + \mathrm{d}t, t\right)}. \tag{10}$$

For our numerical simulations we use Krylov-space methods and time steps of $\mathrm{d}t = 0.1$, which we verified to be sufficient to obtain converged results. We fix the tunneling constant to $J = 1$, which determines the units of time, and we set the noise strength to $\gamma = 1$. We have seen that changing the amplitude of the noise, does not change our results qualitatively. We average our results over 100 disorder realizations and 10 realizations of the noise for each disorder realization. The averages over disorder are denoted by $[\cdot]$ and over the noise by an overbar.

## III. RESULTS

In this section we characterize how the system approaches an infinite-temperature state in terms of the energy and a properly defined entanglement entropy. We then assess the linear response particle transport at infinite temperature.

### A. Energy dissipation

The energy of the system,

$$\varepsilon\left(t\right) = \mathrm{Tr}\left(\hat{\rho}\left(t\right) \hat{H}_A\right), \tag{11}$$

grows as a result of coupling to the local heat bath. Since $\hat{H}_A = \sum_{ij}\left\langle i \left| \hat{h}_A \right| j\right\rangle \hat{c}_i^\dagger \hat{c}_j$, we can express the energy

growth using the single-particle density matrix as,

$$\varepsilon(t) = \sum_{i,j} \left\langle i \left| \hat{h}_A \right| j \right\rangle \mathrm{Tr} \left( \hat{\rho}(t) \hat{c}_i^\dagger \hat{c}_j \right)$$
$$= \sum_{i,j} \rho_{ij}^s(t) \left\langle i \left| \hat{h}_A \right| j \right\rangle, \tag{12}$$

where $|i\rangle$ and $|j\rangle$ are single-particle states in the position basis. Following the discussion of Sec. II, at long times, the system approaches an infinite-temperature state, $\hat{\rho}_\infty \propto \mathbb{1}$, therefore,

$$\varepsilon(t \to \infty) = \frac{1}{\mathcal{N}} \mathrm{Tr} \left( \hat{H}_A \right) = \frac{1}{2} \mathrm{Tr}\, \hat{h}_A = \frac{1}{2} \sum_i^L w_i, \tag{13}$$

where $\mathcal{N}$ is the Hilbert space dimension. Since the energy of the system is bounded, to have a sufficiently wide range of energy growth, we prepare the system in the ground state of $\hat{H}_A$. In this state, the single-particle density matrix is given by,

$$\rho_{ij}^s(0) = \sum_{\alpha=1}^N \phi_\alpha^*(i)\, \phi_\alpha(j), \tag{14}$$

where $|\alpha\rangle$ are single-particle eigenstates of $\hat{h}_A$ and $N$ is the number of fermions, which we set to be $N = L/2$, namely, half-filling.

In Fig. 1(a) we show how the averaged (over realizations of disorder and trajectories) energy absorbed from the coupling to the local environment, $\Delta\varepsilon(t) = \overline{\left[\varepsilon(t) - \varepsilon_{\mathrm{GS}}\right]}$, grows in time for several disorder strengths $W$ and a fixed system size $L = 100$. We observe that the averaged absorbed energy grows logarithmically with time, $\Delta\varepsilon \sim \ln t$, over a broad time window extending into several decades and for all the disorder strengths we study. Increasing the disorder strength further suppresses the heating. Interestingly, this logarithmically slow energy growth resembles the heating in a vicinity of the Floquet-MBL transition [41].

In Fig. 1(b), we show how $\Delta\varepsilon(t)$ depends on system size $L$, for a weak disorder $W = 1$. For the smallest system size $L = 12$, the absorbed energy saturates, but as we increase the system size, the times required to observe saturation significantly increase. Since the energy of the system is extensive, to compare the saturation of the energy for different system sizes we calculate the energy density $\varepsilon(t)/L$. The results are presented in the inset of Fig. 1(b) and show the approach to an energy density corresponding to an infinite-temperature state, which for $\hat{H}_A$ is $\varepsilon_\infty/L = \frac{1}{2}\langle w_i \rangle = 0$.

### B. Entanglement entropy growth

The entanglement entropy is not well defined for mixed states [42], and therefore is not a natural quantity to

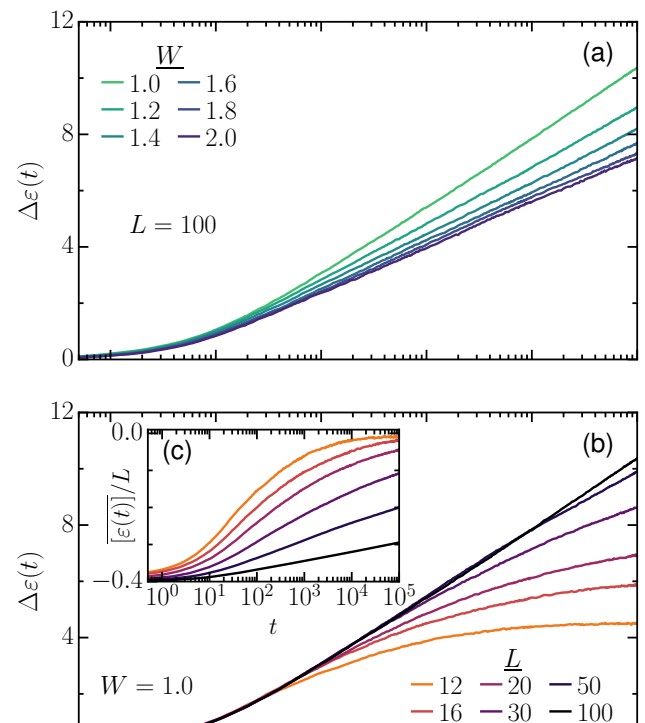

Figure 1. Averaged energy growth. (a) As a function of disorder strength $W$ for $L = 100$, and (b) as a function of system size $L$ for $W = 1.0$.

consider for dissipative dynamics given by (4). However, since (4) can be unraveled into a unitary evolution of an ensemble of pure states (see Sec. II), a well-defined entanglement entropy can be computed for each of the members of the ensemble *separately*, and then averaged. Since the entanglement entropy is not a linear function of the wavefunctions, the averaged entanglement entropies corresponding to different unravelings of (4) do *not* need to coincide [43]. Notwithstanding, the averaged entanglement can obtain meaning, if a certain unraveling can be physically motivated [37]. Specifically, the unitary unraveling of Refs. [39, 40] that we use here, can be thought of as a time-dependent, multi-frequency local potential, with a band-width much larger than any other energy scale.

To calculate the von-Neumann entanglement entropy we partition the system into two spatially equal parts $A$ and $B$. For noninteracting systems the entanglement between $A$ and $B$ is given by [44]

$$S(t) = -\sum_\alpha \left[ \tilde{n}_\alpha^A(t) \log \tilde{n}_\alpha^A(t) \right.$$
$$\left. + \left(1 - \tilde{n}_\alpha^A(t)\right) \log \left(1 - \tilde{n}_\alpha^A(t)\right) \right], \tag{15}$$

where $\tilde{n}_\alpha^A(t)$ are the eigenvalues of the single-particle den-

sity matrix $\rho_{ij}^s(t)$ with $i, j \in A$. To have a sizable regime of entanglement growth we initiate the system at a random product state, namely, $\rho_{ij}^s(t = 0) = n_j \delta_{ij}$, with random $n_j \in \{0, 1\}$.

In Fig. 2(a), we show the time evolution of the entanglement entropy $S(t)$, averaged over disorder and trajectories realizations, for various disorder strengths $W$, and a fixed system size $L = 100$. Similar to the energy, we find that the averaged entropy grows logarithmically with time and that the slope of its growth is suppressed with disorder strength. Since the entanglement is bounded, the growth saturates at long times to $S_\infty \equiv \lim_{t \to \infty} S(t)$, as is apparent for the smaller system sizes in Fig. 2(b). The entanglement entropy density $S(t)/L$, for different system sizes approaches the same constant, $S_\infty/L \approx \frac{1}{4} \ln 2$, as is shown in the inset of Fig. 2(b). This value is considerably smaller than the Page value $S_{\text{Page}} = \frac{1}{2}L \ln 2 - \frac{1}{2}$ [45], contrary to the case of coupling to noise in *interacting* systems [46]. Since in our case the state of the system is a product state for all times, though not necessarily in the position basis, the saturation value better agrees with the entanglement entropy density averaged over all possible product states, given by $\bar{S}/L \approx 0.193$ (see Eq. (2) in Ref. [47]), and not over all possible states in the entire Hilbert space, which would correspond to the Page value. Before concluding this section, it is worthwhile to observe that the entire behavior of the averaged entanglement entropy in Fig. 2 is somewhat reminiscent of the entanglement entropy behavior in many-body localized systems [48, 49], though, as we will see in what follows, here the system is delocalized by the noise, which induces a slow particle transport.

### C. Particle transport at infinite temperature

In the previous subsections we have studied the approach of the system to the infinite-temperature state. Here, we consider particle transport at infinite temperature. Since in the absence of noise, all single-particle states are localized, there is no transport, even at infinite temperature. Therefore, all transport is induced by the local noise. To assess particle transport we compute the density-density correlation function

$$C_{ij}(t) = \text{Tr}\left(\hat{\rho}_\infty \, \hat{n}_i(t) \, \hat{n}_j\right) = \left|\text{Tr}\left(\hat{\rho}_\infty \, \hat{c}_i^\dagger(t) \, \hat{c}_j\right)\right|^2, \quad (16)$$

which corresponds to the spreading of an excitation of the density at site $j$. The last equality follows from the fact that the particles are noninteracting, and $\hat{\rho}_\infty \propto \mathbb{1}$. For the unitary unraveling of (4) that we use here, the evolution of $\hat{c}_i^\dagger(t)$ is given by $\hat{c}_i^\dagger(t) = \sum_k U_{ik}^s(t, 0) \, \hat{c}_k^\dagger$,

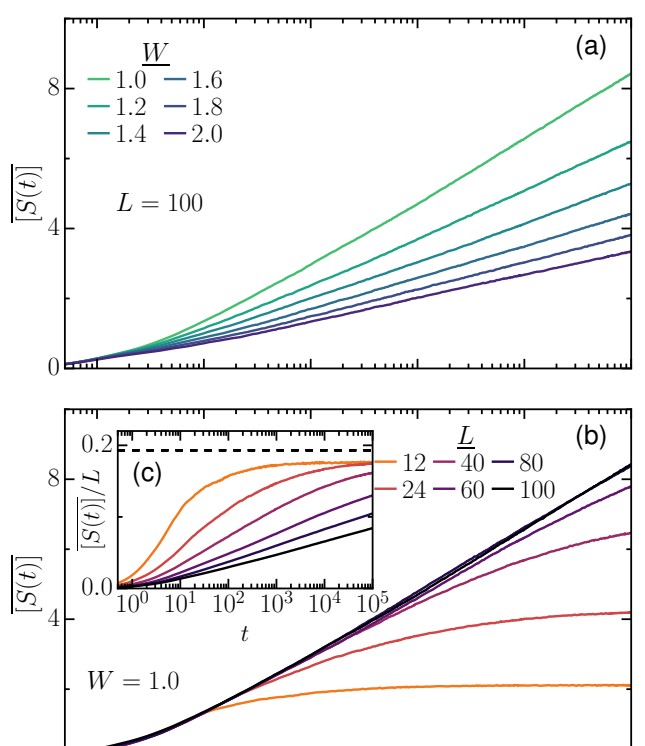

Figure 2. Averaged entanglement growth. (a) As a function of disorder strength $W$ for $L = 100$, and (b) as a function of system size $L$ for $W = 1.0$. (c) Inset showing the saturation of the averaged entanglement entropy density. The dashed line in (c) refers to the entanglement entropy density averaged over all possible product states, $\bar{S}/L \approx 0.193$ (see Eq. (2) in Ref. [47]).

therefore,

$$C_{ij}(t) = \left|\sum_k U_{ik}^s(t, 0) \, \rho_{kj}^s\right|^2 = \frac{1}{4} \left|U_{ij}^s(t, 0)\right|^2, \quad (17)$$

where we used the infinite-temperature form of the single-particle density matrix, $\rho_{kl}^s = \frac{1}{2}\delta_{kl}$. To characterize the nature of transport in the presence of the local noise, we first evaluate the width of the excitation profile, known as the root-mean-square (RMS) displacement,

$$\tilde{R}(t) = \left(\sum_{i=1}^L (i - j)^2 \, \overline{[C_{ij}(t)]}\right)^{1/2}. \quad (18)$$

For diffusive transport, $\tilde{R}(t) \sim \sqrt{2Dt}$, where $D$ is the linear response diffusion constant [50, 51], and for localization the width is bounded, $\tilde{R}(t) \leq A$.

In Ref. [26], it was shown that the Anderson insulator subject to *global* noise with arbitrary correlation time exhibits transient subdiffusion, before asymptotic diffusion takes place. In contrast, in the case of local white

noise, we find that the RMS displacement $\tilde{R}(t)$ grows logarithmically with time, without any signs of crossover to diffusion (see Fig. 3(a)). Similarly to the energy and the entanglement entropy, transport is suppressed with increasing the disorder strength. In Fig. (3)(b) we show that our results do not suffer from finite-size effects over a broad time window spanning several decades.

## IV.  SEMI-ANALYTICAL PICTURE

In this section we provide a theoretical model of nonequilibrium dynamics in the Anderson insulator in the presence of a local noise, which gives a qualitative explanation of the phenomenology we observe. For this purpose we use a variable-range-hopping-like approach, as originally introduced by Mott in the context of phonons [18]. In this approach, the environment induces hopping between the localized orbitals of the Anderson problem. Moreover, it is assumed that the noise decoheres the dynamics, such that the process can be described by a classical master equation [22, 23, 52],

$$\partial_t p_\alpha = \sum_\beta \left( \Gamma_{\alpha\beta} p_\beta - \Gamma_{\beta\alpha} p_\alpha \right), \tag{19}$$

where $p_\alpha$ is the probability to find a particle at an Anderson eigenstate $\langle i | \alpha \rangle = \phi_\alpha(i)$, and

$$\begin{aligned} \Gamma_{\alpha\beta} = \Gamma_{\beta\alpha} &= \gamma^2 \left| \langle \beta | L/2 \rangle \langle L/2 | \alpha \rangle \right|^2 \\ &= \gamma^2 \left| \phi_\beta^* \left( \frac{L}{2} \right) \phi_\alpha \left( \frac{L}{2} \right) \right|^2, \end{aligned} \tag{20}$$

are the transition rates between Anderson eigenstates $|\alpha\rangle$ and $|\beta\rangle$, where to obtain the rates we used the noise coupling $\gamma |L/2\rangle \langle L/2|$, and the fact that the noise is white. Since the Anderson eigenstates are localized, in a one-dimensional lattice the indices $\alpha$ can be ordered almost in one-to-one correspondence with the site indices $i$, thus we can write,

$$\Gamma_{\alpha\beta} = \gamma^2 e^{-|\alpha - L/2|/\xi} e^{-|\beta - L/2|/\xi}, \tag{21}$$

where $\xi$ is the localization length. We see that the transition rates between a pair of states are exponentially suppressed with the distance from the local noise, which explains the exponentially long-time scales we observe. To see this more precisely, we numerically solve (19) for a particle initially located at the center of the lattice. In this case the RMS displacement is given by $\tilde{R}(t) = \sqrt{\sum_\alpha \left( \alpha - \frac{L}{2} \right)^2 p_\alpha(t)}$, and is plotted in Fig. 4(a) for various localization lengths, $\xi$. Plotting $\tilde{R}(t)/\xi$ with respect to $\xi t$ results in a perfect collapse of the data, as shown in Fig. 4(b), indicating that the RMS displacement scales as $\tilde{R}(t) \sim \xi \ln(\xi t)$, which is in excellent agreement with the quantum simulation in the previous

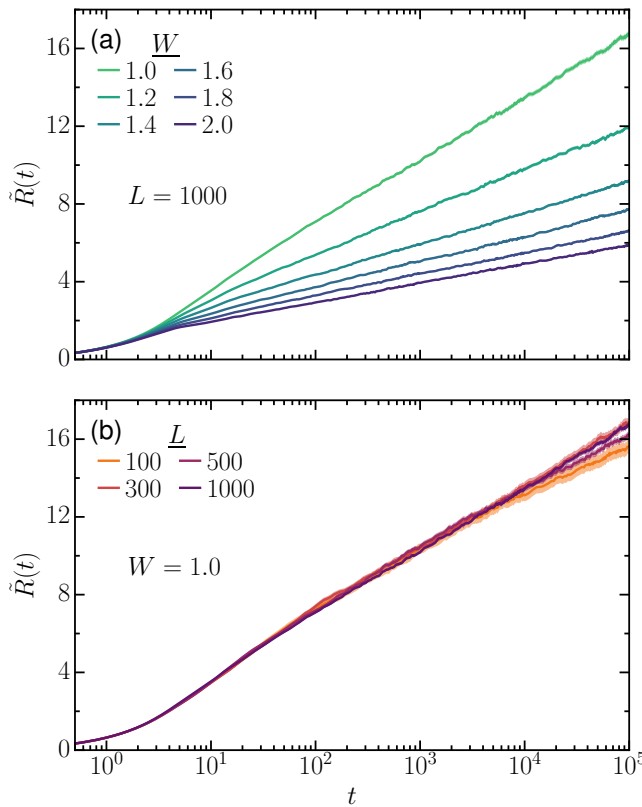

Figure 3. RMS displacement $\tilde{R}(t)$ as a function of time. (a) For various disorder strengths $W$ and $L = 1000$, and (b) for various system sizes $L$ and $W = 1.0$.

section, though we could not produce a similar collapse for the original problem (2) with $\xi$ computed numerically. This might indicate that more than one scaling parameter might be required.

## V.  DISCUSSION

We have studied the dynamical behavior of the Anderson insulator in the presence of a local noisy potential. While the dynamics is dissipative, it can be efficiently studied using an ensemble of pure states, which evolve under *unitary* evolution [39, 40]. Physically, this corresponds to dynamics in the presence of a local, time-dependent potential with a very wide bandwidth of frequencies, which allows us to consider, in addition to the energy absorption, the growth of the entanglement entropy. We find that both quantities grow logarithmically in time and saturate after times that diverge exponentially with system size. While the local noise leads to an infinite-temperature state at long times, the entanglement entropy saturates to an extensive value which is smaller than the Page value [45], but that is in good agreement with the average entanglement entropy over all product states [47]. Interestingly, the entanglement

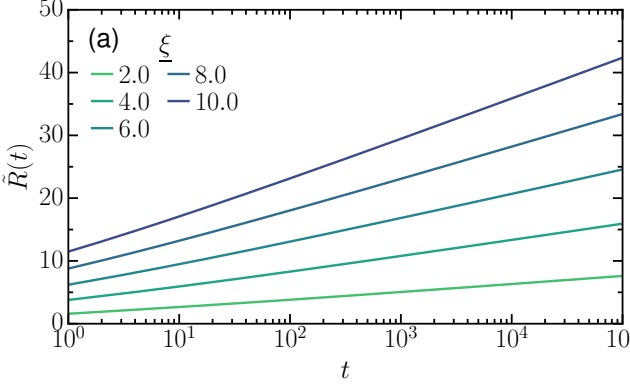

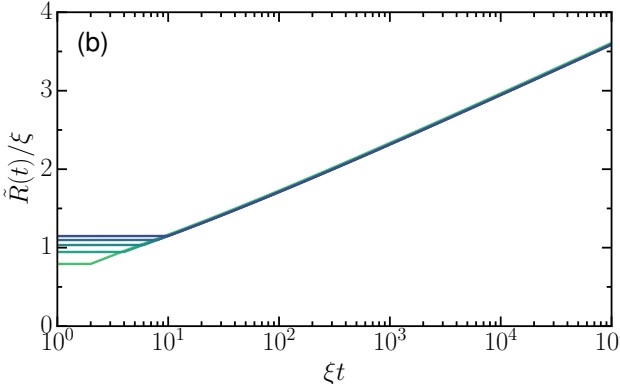

Figure 4. RMS displacement $\tilde{R}(t)$ as a function of time. (a) For various localization lengths $\xi$ and $L = 1000$. (b) The same as (a), but $\tilde{R}(t)/\xi$ plotted vs $\xi t$.

entropy growth we observe is similar to that of many-body localized systems [48, 49], but contrary to MBL systems the local noise induces slow logarithmic particle transport at infinite temperature. This scenario is also different from the case of global coupling to noise, where subdiffusive transport is only a transient and asymptotically the system is diffusive [26].

We show that the slow dynamical behavior of the Anderson insulator in the presence of a local noise, can be qualitatively understood using a classical master equation, which describes noise-mediated hopping of a particle between localized single-particle states, similar to the variable-range-hopping mechanism [18, 22, 23, 52]. In particular, we show that the RMS displacement of the particle grows as, $\tilde{R}(t) \sim \xi \ln \xi t$, indicating a vanishing diffusion coefficient. Based on this analysis, it is easy to see that our results should hold for any noise which operates in a bounded spatial region, however, the system will become delocalized if the noise operates on a finite fraction of the lattice, $p$. In this case, the average distance between the noisy sites would be $\ell = 1/p$, and the system would delocalize in a time scale, $t \sim \xi^{-1} \exp[\ell/\xi]$, exhibiting diffusive transport [26, 34]. Our results provide an upper bound for the delocalization rate of Anderson and MBL systems in the presence of local ergodic grains, discussed in Refs. [53, 54], since unlike the grains the local noise does not "cool down".

We thank Lev Vidmar for bringing to our attention Ref. [47], where the average entanglement entropy over all possible product states is obtained analytically. This research was supported by a grant from the United States-Israel Binational Foundation (BSF, Grant No. 2019644), Jerusalem, Israel, and the United States National Science Foundation (NSF, Grant No. DMR-1936006), and by the Israel Science Foundation (grants No. 527/19 and 218/19). TLML acknowledges funding from the Kreitman fellowship.

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
