# Peer review of "Logarithmic, noise-induced dynamics in the Anderson insulator"

_SciPost Physics_

## Round 2 · Referee Report · Anonymous · 2021-11-2

Strengths

1- The Authors discuss an interesting problem, bridging the gap between global and local noise in the Anderson insulator.
2- The Authors present a comprehensive picture of the model studied.
3- An analytical picture is provided, which explains qualitatively the results obtained numerically.
4- The quality of the presentation is very good, making the paper easy to follow and overall enjoyable.

Weaknesses

1- In some parts of the text, more specification would be beneficial to the reader (see more detailed list below).
2- Generally, the captions of Figures could be improved, adding more details about what the plots show and some interpretation of the data.

Report

In this paper, the Authors address the interesting phenomenon of an Anderson insulator locally coupled to a white noise. While it is known in other regimes that the coupling to Markovian noise leads eventually to diffusive transport, the authors provide evidence that this is not the case when the noise acts locally. The numerical results presented show indeed logarithmic dynamics of many measurable quantities. An analytic picture is also developed, ascribing the logarithmic behavior to the exponentially localized nature of the single-particle wave-functions. Through this picture, the Authors are also able to qualitatively reproduce the results obtained numerically for the RMS displacement.

The work of the Authors provides a comprehensive study of this particular model, which is of interest for the understanding of the localization-environment interplay. The Authors' contribution is original and well presented, hence I recommend its publication in the journal.
Nevertheless, there are some questions I would like to ask the Authors, hoping to stimulate further interesting results.

1- Are there experimental setups that could be used to study this model? In case, it would be interesting to compare numerical and experimental results.
2- Did the Authors consider exploring other noise strengths $\gamma$? Is the presence of a critical strength after which transport becomes diffusive expected?
3- How are the results shown in the manuscript sensitive to changes of initial states? In particular, the logarithmic growth of the energy absorption and entanglement entropy are obtained starting from different initial states. While it is clear to me the rationale behind the choice of the initial conditions, I am curious about the stability of logarithmic growth e.g. when energy dissipation is studied starting from random product states.
4- Did the Authors consider the study of more local probes? It would be interesting to observe the effect of the local noise at different positions in the chain, e.g. the entanglement profile $S(i)$ at some late time. This could give access to how the distance from the noise source affects the system locally.
5- Can the Authors comment on the accuracy of the unraveling procedure depending on the number of trajectory realizations and on the time-step dt?

Requested changes

1- In Eq.(5) the Authors define $\eta_t$, but they do not specify in what interval it is distributed. I think this information should be added, in order to make the results of the Authors more reproducible.
2- In Eq.(14), perhaps it could be helpful to mention that $|\alpha\rangle$ are the $N$ lowest eigenstates of the single-particle Hamiltonian (if that is the case, as I think).
3- The wave-functions $\phi_\alpha(i)$ are defined after Eq.(19), however they are already used in Eq.(14), so I suggest the Authors to define them earlier.
4- In Eq.(18), the Authors calculate $\overline{[\;\cdot\;]}$, instead of $[\overline{\;\cdot\;}]$, is there a reason for this choice, or is it simply a typo?
5- In Eq.(21), I think the Authors are missing the normalization factor of the wave-functions, which would contribute $\approx 1/\xi^2$. Also, there is a small typo,$||$ instead of $|$, in the exponent of the second exponential.
6- I suggest the Authors to specify in the caption of Figure 4 that the results shown there correspond to the analytic picture.

  • validity: high
  • significance: good
  • originality: high
  • clarity: high
  • formatting: excellent
  • grammar: excellent

Author:  Talía L. M. Lezama  on 2022-01-06  [id 2072]

(in reply to Report 1 on 2021-11-02)
Category:
answer to question

We thank the Referee for their careful reading of the manuscript, for their positive comments, and constructive report. We have modified the manuscript based on their suggestions and reply to their comments below.

1-Are there experimental setups that could be used to study this model? In case, it would be interesting to compare numerical and experimental results.

Reply: Indeed, this is an important point. We added a sentence in the introduction and the discussion about the possible experimental setups that could be used to confirm our theoretical results.

2-Did the Authors consider exploring other noise strengths? Is the presence of a critical strength after which transport becomes diffusive expected?

Reply: Yes, we explored different noise strengths. Transport never becomes diffusive, increasing the noise strength only delays transport or relaxation in the system, which is always logarithmic. We attach numerical evidence showing the delay of the logaritmic dynamics with increasing noise strength; (a) shows the entanglement growth and (b) the root mean displacement.

3- How are the results shown in the manuscript sensitive to changes of initial states? In particular, the logarithmic growth of the energy absorption and entanglement entropy are obtained starting from different initial states. While it is clear to me the rationale behind the choice of the initial conditions, I am curious about the stability of logarithmic growth e.g. when energy dissipation is studied starting from random product states.

Reply: The logarithmic growth is stable to the choice of initial states. In the case of the energy, we choose as initial state the ground state to be able to observe the energy growth for several decades before it reaches its saturation value. It is not possible to observe such energy growth when starting from random product states, since the energy in such states is already too close to its saturation value.

4- Did the Authors consider the study of more local probes? It would be interesting to observe the effect of the local noise at different positions in the chain, e.g. the entanglement profile S(i) at some late time. This could give access to how the distance from the noise source affects the system locally.

Reply: We thank the Referee for this suggestion. We have included results for the entanglement profile in the lower panel of Figure 3., showing that the entanglement growth gets delayed with the distance between the cut and the local noise. This is in agreement with our semi-analytical picture where the transition rates between pair of localized states are exponentially suppressed with the distance between their centers of localization and the local noise, expressed in Eq. (21) of the manuscript.

5- Can the Authors comment on the accuracy of the unraveling procedure depending on the number of trajectory realizations and on the time-step dt?

Reply: Increasing the number of realizations increases the accuracy of the unraveling procedure, by suppressing the statistical noise. The advantage of this particular unraveling is that it is exact for any dt, provided the Lindblad operators are Hermitian, as it is our case. Therefore, our results are invariant with respect to dt and don't require dt\to0.

Requested changes:

1- In Eq.(5) the Authors define \eta\left(t\right) , but they do not specify in what interval it is distributed. I think this information should be added, in order to make the results of the Authors more reproducible.

Reply: Done.

2- In Eq.(14), perhaps it could be helpful to mention that \ket{\alpha} are the N lowest eigenstates of the single-particle Hamiltonian (if that is the case, as I think).

Reply: Done.

3- The wave-functions \phi_{\alpha}\left(i\right) are defined after Eq.(19), however they are already used in Eq.(14), so I suggest the Authors to define them earlier.

Reply: Done.

4- In Eq.(18), the Authors calculate \overline{[\cdot]} instead of [\overline{\cdot}], is there a reason for this choice, or is it simply a typo?

Reply: Since we average over both disorder and noise realizations, interchanging the averages yields the same results, however, we meant to write [\overline{\cdot}], which is the correct theoretical expression, so we have replaced \overline{[\cdot]} with [\overline{\cdot}] everywhere.

5- In Eq.(21), I think the Authors are missing the normalization factor of the wave-functions, which would contribute 1/\xi^{2}. Also, there is a small typo,|| instead of |, in the exponent of the second exponential.

Reply: The exponential form in Eq. (21) is correct. The squared absolute value enters as a constant that is absorbed by the localization length \xi, and therefore the contribution should be of the form \exp(-x/\xi). We thank the Referee for pointing out the typo || in the exponent.

6- I suggest the Authors to specify in the caption of Figure 4 that the results shown there correspond to the analytic picture.

Reply: We modified the caption accordingly as well as improved the captions of the other figures.

Attachment:

NoiseStrength.pdf

---

## Round 2 · Referee Report · Anonymous · 2021-11-23

Report

The authors study the effect of local white noise on the one-dimensional Anderson model. In particular, they study the approach to the infinite temperature state starting from the ground state of the Anderson model. They do so by monitoring the time dependence of energy and entanglement of the unraveled dissipative dynamics. They show that both quantities display logarithmic growth. They also demonstrate sub-diffusive transport when the infinite temperature state is reached.
To prove the abovementioned results, they perform numerical simulations for solving the open quantum dynamics. Moreover, they provide a semi-analytical framework that captures the essential features of the phenomenology they observed.
The paper is well written and I believe it meets the criteria for publication on SciPost Physics. I only have a couple of concerns that I think the authors should address in the manuscript.
1) The fact that the system is one-dimensional is not explicitly stated neither in the title nor in the abstract. I think this should be made clear from the very beginning since it is a strong limitation on the generality of their results. If the authors believe this is not the case, they should discuss how their results extend to more than one dimension.
2) As the authors stress in the manuscript, the entanglement entropy of mixed state is not a good entanglement measure (they actually say that it is not well defined, a statement that I would change with what I just wrote). They proceed anyway to the calculation of the entanglement entropy of the unitary unraveled dynamics, arguing that such quantity has a meaning, given the physical interpretation of the unraveling they use. I think the authors should elaborate a bit more on this. More specifically, I fail to see the implications of the log(t) scaling they found for this entanglement entropy, and their physical interpretation. I would have also liked to see a discussion of the dependence of this scaling from the unraveling adopted.

  • validity: good
  • significance: ok
  • originality: good
  • clarity: high
  • formatting: good
  • grammar: excellent

Author:  Talía L. M. Lezama  on 2022-01-06  [id 2073]

(in reply to Report 2 on 2021-11-23)
Category:
answer to question

We thank the Referee for their careful reading of the manuscript, positive comments, and constructive report. We address the requests made by the Referee and reply to their comments below.

1) The fact that the system is one-dimensional is not explicitly stated neither in the title nor in the abstract. I think this should be made clear from the very beginning since it is a strong limitation on the generality of their results. If the authors believe this is not the case, they should discuss how their results extend to more than one dimension.

Reply: We have specified that the system is one-dimensional in the abstract. While we cannot state with certainty that the same behavior will generalize to higher dimensions, the presented analytical picture strongly suggests it is the case, at least in the absence of delocalized single-particle states. We have clarified this point in the text.

2) As the authors stress in the manuscript, the entanglement entropy of mixed state is not a good entanglement measure (they actually say that it is not well defined, a statement that I would change with what I just wrote). They proceed anyway to the calculation of the entanglement entropy of the unitary unraveled dynamics, arguing that such quantity has a meaning, given the physical interpretation of the unraveling they use. I think the authors should elaborate a bit more on this. More specifically, I fail to see the implications of the log(t) scaling they found for this entanglement entropy, and their physical interpretation. I would have also liked to see a discussion of the dependence of this scaling from the unraveling adopted.

Reply: The Referee is right, this part of the manuscript was a bit confusing. We have replaced ``[i]s not well defined" by "is not a good entanglement measure" and modified the paragraph explaining more clearly what we meant. We also included a new paragraph expressing the implications of the log(t) scaling and the physical interpretation. When the unraveling procedure can be physically motivated, the averaged entanglement bounds from above spreading of correlations and transport in the system. The dependence of the entanglement on the unraveling used is a complicated subject on its own, this is why we decided to refer the reader to the appropriate literature.

---

## Editorial Decision

resubmitted